# Phenotypic and Genetic Study of the Presence of Hair Whorls in Pura Raza Español Horses

**DOI:** 10.3390/ani13182943

**Published:** 2023-09-16

**Authors:** Ana Encina, Manuel Ligero, María José Sánchez-Guerrero, Arancha Rodríguez-Sainz de los Terreros, Ester Bartolomé, Mercedes Valera

**Affiliations:** 1Real Asociación Nacional de Criadores de Caballos de Pura Raza Española (ANCCE), 41014 Sevilla, Spain; arancha@lgancce.com; 2Departamento de Agronomía, Escuela Técnica Superior de Ingeniaría Agronómica, Universidad de Sevilla, Carretera de Utrera Km 1, 41013 Sevilla, Spain; manligaco1999@gmail.com (M.L.); msanchez73@us.es (M.J.S.-G.); ebartolome@us.es (E.B.); mvalera@us.es (M.V.)

**Keywords:** behavior, heritability, equines, coat color, genetic models, temperament

## Abstract

**Simple Summary:**

Hair whorls in horses are a hereditary trait that may be associated with various factors, including the temperament or the coat color of the animals. Hair whorls are described as changes in the hair pattern and may take various forms: circular whorls (spirals of hair with a round epicenter) and linear whorls (a line where the hairs span out on both sides from the center, producing an oval shape similar to a feather). The aim of this study is to estimate the frequency and genetic parameters of the number and position of circular and linear hair whorls (on head, body-neck and limbs) of the Pura Raza Española horse according to different factors such as gender, level of inbreeding, birth period and coat color. In this breed, circular whorls are more prevalent than linear whorls, with both showing a relevant symmetry. The laterality of hair whorls has been also evidenced and are most concentrated on the left-hand side. Most horses, particularly gray ones, showed circular hair whorls below the central line of the eyes; in a previous paper, this was associated with a calmer and more docile temperament. Hair whorls have medium-high heritability and can be included in a breeding program due to their relationship with behavior.

**Abstract:**

Hair whorls are a hereditary feature in horses that may be associated with temperament and coat color. Hair whorls are described as changes in the hair pattern and may take various forms, such as circular and linear whorls. We first carried out a frequency analysis of hair whorls (circular and linear). Next, a Generalized Non-Linear Model was computed to assess the significance of some potential influencing factors, and a genetic parameter estimation was performed. ENDOG software v4.8 was used to estimate the inbreeding coefficient of all the animals analyzed. It was more common to find horses with circular hair whorls than with linear whorls. The heritability ranges obtained were, in general, medium-high for both circular whorls (0.20 to 0.90) and linear whorls (0.44 to 0.84). High positive correlations were found on the between left and right positions, indicating a tendency to symmetry in certain locations. The laterality of hair whorls was also evidenced, with the biggest concentration on the left-hand side, particularly in gray horses, showing circular whorls below the central line of eyes, which has been associated in a previous paper with a calmer and more docile temperament.

## 1. Introduction

Numerous papers have studied the relationship between body conformation, coat color or coat particularities and temperament traits in domestic animals [1,2,3,4,5,6,7,8]. Among the particularities which have been studied in greater depth are hair whorl patterns [5,9,10,11,12]. These hair features, also called trichoglyphs, are laid down between the 10th and the 16th weeks in utero in humans and can act as an indicator of abnormal cerebral development [13]. It is widely known that hair whorls come from the same layer of cells in the embryo as the nervous system [14], and trichoglyphs have been linked to early fetal brain development. Further study of hair whorls may therefore provide useful insight into both behavioral and neurobiological development [15], having been related in humans to neuro developmental disorders such as schizophrenia. In addition, in dogs, relationships have been found between hair whorls and motor and sensory laterality [16,17].

In this context, the existence of an asymmetric connection between brain function and behavior has also been previously demonstrated [18,19]. In animals, the right hemisphere is related to flight response before new events, while the left hemisphere controls routine behavior when the animal is relaxed [16,20,21].

In horses, since ancient times, many superstitions have been related to the number and position of hair whorls, as pointed out by Hudayl [22]; for example, hair whorls that appear on the neck and in the interaxillary region are considered good omens, and those on the lower part of the thigh, and on the jaw, back and shoulders bring bad luck. Aguilar [23], writing in 1572, considered it a bad sign that hair whorls were located on the temples, jaws, back or heart. At the beginning of the 20th century, Suë [24] referred to the Arabian horse “Godolphin”, one of the founders of the thoroughbred horse, as an animal with a hair whorl on its chest that “boded ill for its owner”.

However, nowadays, this association between the presence of hair whorls, mainly on the forehead, and equine temperament has been scientifically studied, and the results have been presented in different papers [9,12,25].

Certain genomic regions associated with hair traits in horses have now been identified [26]. The possible pleiotropic effect of these genes, whose metabolic activity is related to hair follicle growth, has been also highlighted, thus providing the first evidence of a genetic–biological association between hair whorls and temperament in horses [26]. In general, the phenotype of hair whorl traits is a good indicator of the genotype, since these traits have a large additive genetic variance. In addition, hair whorl traits in horses show high heritability [27,28,29] and are highly correlated with one another [27]. On the other hand, other particularities of whorls, such as linear whorls, have not been studied in depth before in horses.

The Pura Raza Española (PRE) horse is the main native Spanish equine breed, with a census of 275,018 animals, with 26.23% of its census being distributed in 67 countries spread over the five continents. It is a breed that has contributed to the formation of numerous equine breeds [30], and the horses of this breed stand out for being balanced, docile and with a calm temperament. They also adapt with ease to different situations, such as working in the field or in equestrian competitions such as Dressage or High School Dressage [31]. They are horses that respond easily to the rider’s encouragement, which is why they are considered obedient, and they have a great capacity for learning.

The aim of this study is to estimate the frequency and genetic parameters (heritability and genetics correlations) of the number and position of circular and linear hair whorls (on head, body–neck and limbs) of the PRE horse according to different factors such as gender, level of inbreeding, birth period and the horses’ coat color.

## 2. Materials and Methods

### 2.1. Description of the Traits and Database

The database included records of 43,398 PRE horses with a range of ages from 6 to 12 months (15,782 stallions and 27,616 mares) taken as part of the veterinary identification process between 2000 and 2021. Veterinary identification is a mandatory condition for registering a horse in the Birth Registry of the PRE-Stud Book and involves carrying out a phenotype review of the animal by an authorized PRE vet. For this purpose, all the reviews from the PRE population are performed by 13 different authorized veterinarians, all of whom are familiar with the identification review routine for this breed. According to the principles of equine identification [32], hair whorls are described as changes in the hair pattern and may take various forms. In this study, we are going to look at circular whorls (spirals of hair with a round epicenter [15]; Figure 1a) and linear whorls (a line where the hairs change their direction on both sides from the center, producing an oval shape similar to a feather).

The position traits were analyzed on a dichotomic scale (with or without circular or linear whorls on a certain position) and on a linear discrete scale (number of whorls in the same location). All possible positions and number of hair whorls situated on head, body-neck and limbs were considered (Table 1). Furthermore, in order to assess the symmetry in the appearance of these traits, we also evaluated the frequency of an animal with a hair whorl on the right side also having one on the left side, for all the locations considered.

### 2.2. Statistical Analyses

First, we performed a frequency analysis of whorls (linear and circular). Next, a Generalized Linear Model (GLZ) was computed to assess the significance of some potential factors that could affect the number or distribution of these hair particularities over the horse body. The GLZ represents an extension of the general linear model designed to accommodate dependent variables with non-continuous or non-normally distributed values. A Generalized Linear Model of the binomial with a logit link function (glm (…, family = binomial (link = “logit”), …)) was used for counts of dichotomous traits; while, for discrete traits, GLZ for an ordinal multinomial distribution (a multinomial discrete distribution containing information on a rank scale) was used. The following factors included in the model were coat color (4 levels: grey, bay, black and chestnut); sex (2 levels: male and female); birth period (3 levels: animals born before 2001, between 2002 and 2011 and between 2012 and 2021); and the inbreeding coefficient (3 levels: <0.125; between 0.125 and 0.1875; and >0.1875), together with its influence on the number and position of hair whorls.

The birth period focused on events in the last few decades, especially since the breeding program for the breed was initiated in 2002, and because, in recent years, breeders have selected their animals based on breeding values of morphological and functional traits related to Dressage, where the horse’s behavior is of utmost importance for the good performance of the routines. Additionally, in the last birth period, there has also been an increase in the prevalence of dark coat colors (bay, black, and chestnut [33]) in this breed. For this reason, the birth period helps us see what happened to the animals born in these last birth periods [34]. The inbreeding coefficient has been considered as a fixed factor, establishing a series of classes or levels, as in other studies in this breed [35].

Statistical analyses were performed using Statistics for Windows software v.11 [36].

### 2.3. Genetic Model

The estimation of the genetic parameters was carried out with a multivariate model (two to six traits carried out by each approach) animal model, with the number of hair whorls as linear discrete traits and the positions of hair whorls as dichotomic traits. The approaches included as systematic effects all the influencing factors that resulted statistically significant (*p* < 0.05) in the previous GLZ analysis: gender, the inbreeding coefficient, the coat color and the birth period. Additive genetic and residual effects were included as random factors.

All the genetic models were analyzed using a Bayesian approach via Gibbs sampling with the GIBBSF90+ (http://nce.ads.uga.edu/wiki/doku.php?id=readme.gibbsf90plus#gibbsf90, accessed on 14 June 2023) module of the BLUPF90 software [37]. The Gibbs sampler was run for 250,000 rounds, with the first 10,000 considered as burn-in; then, every 10th sample was saved for later analysis. Posterior means and standard deviations were calculated with the POSTGIBBSF90 software (version 3.15) [37] to obtain estimates of variance components. Convergence of the posterior parameters was assessed by visual inspection of the trace plots of posterior distributions generated by the Coda R package [38]. The equation in matrix notation for the model to be solved for a hypothetical trait considering all of the possible random effects was
*y_i_* = x*_i_*b + z*_i_*u + *e_i_*
where y is the vector of observations, b are vectors of the systematic effects, u are vectors of the additive genetic effects, and x*_i_* and z*_i_* are incidence vectors for systematic and animal effects, respectively. The pedigree information for the genetic evaluation included 95,292 horses and was obtained from the official PRE studbook, reconstructing a minimum of 4 generations from each horse evaluated. ENDOG v4.8 software [39] was used to estimate the inbreeding coefficient of all the animals analyzed.

## 3. Results

### 3.1. Statistical Analysis

We carried out a frequency analysis for both the position and number within the body location of hair whorls (circular and linear) in this breed. As regards the position of the whorls (Table 2), our results showed that it was more common to find horses with circular whorls on any part of the body than with linear whorls. The majority of PREs had circular whorls on the head and body–neck (95%; 41,570 and 41,379 PRE horses, respectively) in contrast with the horses with circular whorls on the limbs (16%; 4621 horses). The position with the highest percentage of PREs with circular whorls was the left side of the body–neck, with 38,757 PREs affected (89.30%). In fact, the left side always had more circular whorls than the right side, except in the hindlimbs. All the locations showed a high percentage of animals with symmetry between both sides, with 87% (20,972 horses) for the head, 99% (38,252 horses) for body–neck and 88% (3855 horses) for limbs. In addition, when the head position traits were grouped into the upper or lower part of the head, most horses showed hair circular whorls under the central line of the eyes.

On the other hand, there was a majority of PREs with circular whorls on the head and body–neck (95%), in contrast with the horses with circular whorls on their limbs (18%). The position with the highest percentage of PREs with circular whorls was the left side of the body–neck, with 38,757 PREs (89.30%). Again, the left side always had more circular whorls than the right side, except in the hindlimbs. All the locations again showed a high percentage of animals with symmetry between both sides, with 53% (22,687 horses) for the head, 88% (38,252 horses) for body–neck and 8% for limbs. As before, when the upper and lower head positions were considered together, most horses showed circular hair whorls above the central line of the eyes, with percentages between 30% (low center) and 47% (low left).

Table 3 shows the results for the number of circular and linear hair whorls a PRE horse has in the same location. Most PRE horses showed one (34%) or three (31%) circular hair whorls on the head, more than two on the body-neck (88%) and two on the limbs (3705 PRE −8%). However, most of the horses studied presented more than 10 linear whorls on the body–neck (1038 −2%).

The Generalized Non-Linear Model (GLZ) and percentage comparative analyses of the influencing factors related with the appearance of linear and circular hair whorls on PRE horses are shown in Appendix A. In general, all the factors studied (gender, inbreeding coefficient, coat color and birth period) were statistically significant (*p* < 0.05).

The gender affected 10 positions of the circular (but not top of the head or center body–neck) and 7 linear whorls, with the stallions in general having more prevalence in both. The inbreeding coefficient affects circular and linear whorls in 5 positions and had a higher frequency in PREs with an inbreeding coefficient of over 0.1875. Coat color was a relevant effect for 12 circular and 10 linear whorl positions, with the gray PRE horse showing the highest frequency. The birth period was a relevant factor for 11 circular whorl and 10 linear whorl positions, with PREs born between 2002 and 2011 highest the prevalence.

Regarding the number of whorls, the gender was relevant in animals for head and limbs in circular and body–neck and limbs in linear whorls, with the males having a higher number than the mares. The inbreeding coefficient affected the number of circular whorls in body–neck for circle and linear whorls traits, with, in general, a higher degree of inbreeding in PREs producing a greater number. Coat color was a significative factor in all circular and linear whorl number traits, with gray horses having the highest number of circular whorls in the three locations. The birth period was a relevant factor for all the circular and linear whorl traits, with PREs born after 2002 and 2011 being those with highest prevalence.

### 3.2. Genetic Parameters

The genetic parameters of the models carried out in this study for body location and position of circular and linear hair whorls are presented in Table 4. The heritability ranges obtained were, in general, medium-high for circular whorls (0.20 to 0.90) and for linear hair whorls (0.44 to 0.84). Within the heritability values obtained for circular whorls, the highest were found on the top right (0.80) and top center (0.85) of the head, in all positions of body and neck (0.83 to 0.89) and on the left hindlimb (0.90). However, for linear whorls, the highest heritability values were found on the lower right-hand side (0.84) of the head, followed by the right hindlimb (0.83).

On the other hand, when considering heritability values for the number of whorls in each location (Table 5), circular hair whorls ranged from 0.32 for limbs to 0.51 for the body–neck, while linear whorls ranged from 0.26 for the head to 0.58 for the body–neck: the latter location showed the highest heritability values for both hair whorls (circular and linear).

Table 6 shows the genetic correlations among the different positions within a location. A total of 31% of the correlations have a medium-high magnitude (≥0.25), with 89% being positive and 11% being negative. High positive correlations were found on the head between the lower left and right positions for both linear (0.99) and circular whorls (0.81), indicating a tendency towards symmetry in these locations. In the same way, symmetry was also found on the limbs, due to the high correlations found between right and left forelimbs for circular (0.93) and for linear whorls (0.93); and the high correlations found between right and left hindlimbs, for both circular (0.83) and linear whorls (0.99). If the laterality is studied, the linear whorls in the lower part of the head have medium correlations with the limbs on the same side, while in circular whorls, this laterality can be observed in close correlations on both sides between the lower part of the head and the body.

Finally, Table 7 shows the genetic correlations between the number of circular and linear hair whorls in different locations. The highest genetic correlations were found between the number of whorls located on the body–neck (0.43), with a genetic tendency to show the same number in both particularities. Finally, a close genetic correlation was found between the number of linear whorls located on the head and limbs (0.45) and linear hair whorls located on the limbs, which tended to be accompanied by linear hair whorls on the head.

## 4. Discussion

Hair normally follows a certain direction, namely from top to bottom, from front to back on the sides and in the opposite direction on the back and lower areas. However, every animal has certain peculiarities in the direction their hair grows that will remain with them for their lifetime. This, as well as the coat color, allows the animal to be correctly identified and aids the veterinary identification process. These particularities can be divided into three general groups: variation in hair direction, which can occur all over the animal’s body; the appearance of white hairs in depigmented skin areas; and primitive marks, which are defined according to the location described in the zootechnical breed standards [40].

With the domestication of the horse, artificial selection by men began fine-tuning the physiognomy and temperament of the modern horse. Those horses that stood out for their docile temperament and resistant conformation were given priority for reproduction. However, coat color was the attribute that attracted the most attention and was first selected by human beings, which led to an increased diversity in coat colors [31]. Horses were domesticated in this way, by giving mating priority to those whose coat colors were more showy [41].

The differences in behavior between animals of the same species and/or breed can be attributed to the temperament of each individual, in which genetics can play a major role. In human–horse relations, generic terms are attributed to horses, based on their owners’ daily experiences and according to the animal’s temperament and reactivity, such as being calm or nervous [25]. However, there are also descriptions of the relationship between phaneroptical characteristics and temperament, such as coat color and its relationship with reactivity [3] or the position of head whorls above or below the line of the eyes supposedly influencing personality characteristics of the animals as being more “calm” or “nervous” [5,9,12,15,42,43]. A number of different theories exist to explain the development of hair whorls, including genetics, molecular mechanisms and mechanical tension [44].

The relationship between the height of a hair whorl on the head and temperament has been previously explained due to the hair patterns in the fetus forming at the same time as the brain [13]. Our results showed a higher prevalence of whorls on the lower part of the head (below the center line of the eye) for PRE horses (Table 2), which agrees with the study by Lanier et al. [10] in cattle, who found that this location was related with a calmer and more noble temperament in the animal [10]. Previous studies have found differences in horse breeds according to temperament [45,46], with PRE horses being described as noble, calm and sound [47,48]. Curiously, animals with a grey coat color are the ones that have presented the highest number of whorls on the lower part of the head, both on the right and left sides (see Appendix A). In this context, horses with a gray coat color of the breed Caballo de Deporte Español (CDE) presented lower infrared temperature values at rest, measured with infrared thermography, compared to chestnut, bay or black-coated CDE horses [3]. Other authors [1,4,49] have also reported differences in coat colors related to personality traits such as calmness, boldness or nervousness which, to some extent, are related with the way the animal perceives threat stimuli and reacts by showing stress [4,50,51,52,53]. Therefore, the position and the number of whorls on the head in PRE horses could be used as a possible predictor of temperament from an early age, and top-of-the-head hair whorls could be used as an indicator trait for use in selection against flighty temperaments [5]. In addition, its ease of measurement and stability over time would be an attractive feature if the phenotypic correlation with behavior was confirmed at the genetic level. Due to the fact that hair whorls are formed during embryonic development, this study has found, as might be expected, a high proportion of symmetry.

In our study, the number of circular whorls does not follow a Poisson distribution, as one and three whorls are much more common than two and four. Counting symmetric whorls just once may show more similarity to the theoretical Poisson distribution. In any case, our results indicate that the presence of (some) whorls is closely associated with that of others (whorls are not independent). Specifically, the place where the greatest symmetry occurs is in the body–neck, with 99% and 98% for circular and linear hair whorls, respectively. These results are in line with the theory that vertebrates are assumed to show a left–right symmetric musculoskeletal system [54,55]. In spite of this, and although both sides of the body should genetically have the same characteristics, tending therefore to be symmetrical (such as white spots or hair whorls), the common asymmetry that is usually found in all animals between the left- and right-hand parts of the body could be probably due to variability in cell movements, clonal proliferation of melanoblasts and cell survival [56]. Furthermore, certain non-genetic factors might also play a role in the expression of the position and pattern of the hair whorls [9]. In the case of the PRE horse, the body position with the highest degree of asymmetry was the top of the head.

In our study, we also found differences in the frequency of appearance of hair whorls on the body in PRE horses on the left- or right-hand sides. These differences could be due to the lateralization effect of the brain. The slight increase in the number of left-sided hair whorls (found in the PRE horses mainly on the head and hindlimbs) could be related with a right-handed tendency [12,57]. In the case of PRE horses, this could be related to the tendency to use the right limb most and to turn right during the regular dressage exercises. However, Shivley et al. [12] only found this tendency to be related with the direction of hair whorls (counterclockwise or clockwise).

Although some published papers have addressed the prevalence of hair whorls in horses [12,25,26,27,28,29] or cattle [5,15], no analyses of the environmental factors that may influence the frequency of these peculiarities have been carried out before. In fact, most authors did not include fixed effects in the model when estimating genetic parameters for these variables [27,28]. Only Yokomori et al. [29] included the gender effect in their genetic models.

In our study, we have analyzed the influence of not only gender, but also coat color, inbreeding coefficient and birth period of the horse on the number and location of hair whorls on the horse’s body (Appendix A). In general, all these effects were statistically significant for most of the traits analyzed.

The environmental factors included in this study have already been used previously in this equine breed to estimate genetic parameters related to morphological defects such as cresty neck [58], ewe neck [59] or limbs defects [35]; those related to pathologies such as the presence of skin melanomas [60] or obesity [61]; and those related to the evaluation of morphological [62], behavioral [48], reproductive [63] or performance [64] variables.

As regards heritability results, it should be noted that this study is the first to estimate genetic parameters for each location of hair whorls on the head, neck and body, and limbs as independent variables (not only the general position or number of the whorls), and it is also the first study to take into consideration the number and distribution of linear whorls (Table 4 and Table 5). First of all, the low-high heritability values found for hair whorl positions (0.20 to 0.89) contrasts the medium-low heritability values found for the number of hair whorl traits (0.26 to 0.58). These hair whorl heritability values were therefore in line with those obtained by Cruz et al. [27] and higher than those reported by Yokomori et al. [29].

In general, whorls are highly heritable, and as a result, the number of animals exhibiting them increases over time, which coincides with the fact that horses born in the last birth period (>2012–2021) present a higher proportion of these characteristics (Appendix A), perhaps due to indirect selection (seeking animals with a specific behavior and coat color). Temperament may have been selected divergently between breeds (even indirectly), and this fact may have altered the distribution of the number of facial whorls of the different breeds/populations, with consequent changes in the variance components [27]. Breed is known to have a great influence on horse behavior [65], and behavioral and temperament traits in horses have low to moderate heritability [65,66,67]. Thus, hair whorl traits could potentially be used for indirect temperament selection.

Additionally, some genes have recently been found that act on epidermal biology and control hair follicle growth (genes KLF5, IL2, SIRT1, CD47, CD200, ALDH1, A1), which may play an important role in the formation of whorls and, in turn, have a pleiotropic effect on behavior [26]. In other species, such as Rhodesian Ridgeback dogs, the ridge gene complex (FGF3, FGF4, FGF19, ORA, OV1) was detected in a ROH island on chromosome 18. Genes located within the ROH islands were HMGA2, affecting body size, MSRB3 for floppy ears and MSTN for muscling patterns, besides numerous health related genes, which have been proven to play a role in dog diseases, along with other genes described only in other species [68,69].

In our study, the proportion of linear whorls is low, with more than 90% of horses not having any on their head and limbs, while more than half have them on their body and neck. According to previous studies, linear whorls could be caused by a single mutation and may be associated with diseases, as in the case of Rhodesian Ridgeback dogs [68].

In agreement with our frequency data (Table 2), the genetic correlations between the different locations for the hair whorl (Table 6) were high and positive, mainly on both sides in the same position (right and left) and mainly for the locations of the lower head, body–neck and limbs, which corroborates the results for symmetry found previously in other studies [12]. These findings suggest that the same set of genes coordinates the position of hair whorls on both sides.

## 5. Conclusions

Hair whorls are medium-high heritability traits mainly affected by coat color, gender and birth period and, to a lesser extent, by the levels of consanguinity. While gender and time of birth may causally influence the formation of whorls, coat color likely correlates with hair whorls, rather than influencing them causally. In PREs, circular whorls are more prevalent than linear whorls in all the locations, with both showing a relevant symmetry between the left and right positions. The laterality of hair whorls has been also evidenced, with most of them concentrated on the left-hand side. On the other hand, most horses, particularly gray ones, showed circular hair whorls below the central line of the eyes. Taking into account previous associations of hair whorl position and temperament in cattle, we could infer a relationship between hair whorls and a more docile and calm temperament in horses, usually associated with this horse breed (PRE) and with this coat color (gray) in particular. However, further studies are needed to confirm this hypothesis.

## Figures and Tables

**Figure 1 animals-13-02943-f001:**
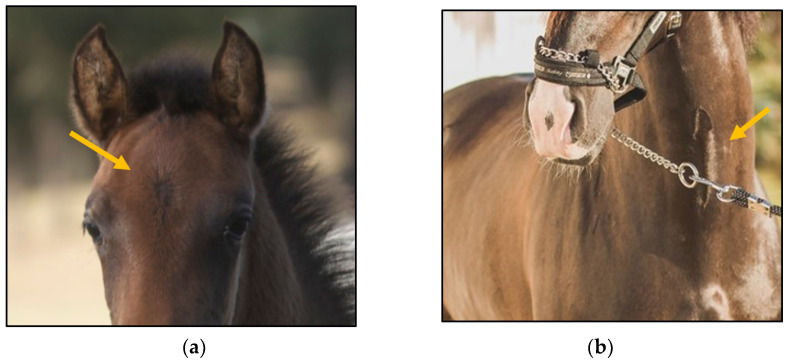
(**a**): circular hair whorl on the center of the head, above the eye line and (**b**) linear hair whorls on the ventral part of the neck. Yellow arrows indicate the location of the circular whorl (**a**) and the linear whorl (**b**).

**Table 1 animals-13-02943-t001:** Position and number of whorls and linear whorls located on the head, body–neck and on the limbs on Pura Raza Español horses.

Location		Position	Number	
**Head**	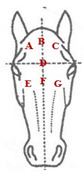		**Circular whorls**	**Linear whorls**
		**A**: top right	**Class 1**: 1	**Class 1**: 1
		**B**: top center	**Class 2**: 2	**Class 2**: 2
		**C**: top left	**Class 3**: 3	**Class 3**: ≥3
		**D**: between the eyes	**Class 4**: 4	
		**E**: lower right	**Class 5**: ≥5	
		**F**: lower center		
		**G**: lower left		
**Body and Neck**	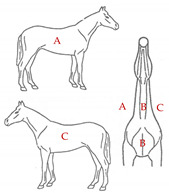		**Class 1**: 1–2	
			**Class 2**: 3–4	
		**A**: right	**Class 3**: 5–6	
		**B**: center	**Class 4**: 7–8	
		**C**: left	**Class 5**: 9–10	
			**Class 6**: >10	
**Limbs**	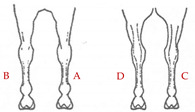	**A**: right forelimb	**Class 1**: 1	
		**B**: left forelimb		
		**C**: right hindlimb	**Class 2**: 2	
		**D**: left hindlimb	**Class 3**: ≥ 3	

**Table 2 animals-13-02943-t002:** Number (above) and percentage (below, in brackets) of the exact body location of circular and linear hair whorl phenotypes in Pura Raza Español horses.

Location		Without	Top Right	Top Center	Top Left	Between Eyes	Low Right	Low Center	Low Left	Symmetry
Head	Circular Whorls	1828 (4.21%)	993 (2.28%)	10,785 (24.85%)	1728 (3.98%)	23,396 (53.91%)	21,430 (49.38%)	8131 (18.74%)	22,308 (47.03%)	20,972 (87.25%)
	Linear Whorls	39,124 (90.15%)	40 (0.09%)	618 (1.42%)	78 (0.18%)	1156 (2.66%)	1501 (3.46%)	1569 (3.62%)	1546 (3.56%)	1422 (87.56%)
Body–neck		Without	Right	Center	Left					Symmetry
	Circular Whorls	2019 (4.65%)	38,712 (89.20%)	37,335 (86.03%)	38,757 (89.30%)					38,252 (98.70%)
	Linear Whorls	16,671 (38.41%)	18,910 (43.57%)	24,325 (56.05%)	18,906 (43.56%)					18,571 (98.21%)
Limbs		Without	Right forelimb	Left forelimb	Right hindlimb	Left hindlimb				Symmetry
	Circular Whorls	38,777 (83.35%)	666 (1.53%)	696 (1.60%)	3723 (8.58%)	3699 (8.52%)				3885 (87.92%)
	Linear Whorls	39,898 (91.94%)	250 (0.58%)	271 (0.62%)	3128 (7.21%)	3100 (7.14%)				3056 (89.91%)

The ‘symmetry’ column indicates that the same feature was found on the left and right side within the area studied. The percentage was calculated taking into consideration the number of horses with left + right whorls in the same location as a percentage of the maximum number affected in this location on the left or right side.

**Table 3 animals-13-02943-t003:** Number (above) and percentage (below, in brackets) of linear and circular hair whorl phenotypes present in each location in Pura Raza Español horses.

		Without	1	2	3	4	≥5	
Head	Circular Whorls	1828 (4.21%)	14,710 (33.89%)	5915 (13.62%)	13,645 (31.44%)	4218 (9.72%)	3086 (7.11%)	
	Linear Whorls	Without	1	2	≥3			
		39,124 (90.15%)	2044 (4.71%)	1398 (3.22%)	319 (0.73%)			
Body–neck		Without	1–2	3–4	5–6	7–8	9–10	>10
	Circular Whorls	2019 (4.65%)	3287 (7.57%)	6907 (15.92%)	8751 (20.16%)	8120 (18.71%)	6002 (13.83%)	8312 (19.15%)
	Linear Whorls	16,671 (38.41%)	7390 (17.03%)	5386 (12.41%)	6736 (15.52%)	4397 (10.13%)	1780 (4.10%)	1038 (2.39%)
Limbs		Without	1	2	≥3			
	Circular Whorls	38,777 (83.35%)	692 (1.59%)	3705 (8.54%)	224 (0.52%)			
	Linear Whorls	39,898 (91.94%)	419 (0.97%)	2942 (6.78%)	139 (0.32%)			

**Table 4 animals-13-02943-t004:** Genetic variance (σ_u_), residual variance (σ_e_) and heritability (h^2^) values for hair whorls according to body location and position.

Location		Position	σ_u_			σ_e_			
			Mean	Median	HPD 95%	Mean	Median	HPD 95%	h^2^
Circular Whorls	Head	Top right	3.958	3.919	3.609–4.378	0.976	0.982	0.926–1.019	0.802
		Top center	5.187	5.161	4.064–6.282	0.933	0.950	0.827–1.010	0.848
		Top left	0.377	0.378	0.301–0.457	1.000	1.000	0.977–1.025	0.274
		Between eyes	0.246	0.244	0.200–0.295	1.017	0.997	0.666–1.383	0.195
		Low right	0.295	0.244	0.085–1.118	1.002	0.999	0.913–1.116	0.227
		Low center	3.362	3.775	0.642–5.161	0.998	0.998	0.904–1.099	0.771
		Low left	0.276	0.206	0.128–0.905	1.001	1.001	0.904–1.079	0.216
	Body-neck	Right	8.011	7.930	5.731–10.240	0.985	0.986	0.958–1.013	0.890
		Center	7.793	7.674	5.733–10.110	0.986	0.986	0.959–1.012	0.880
		Left	4.860	4.853	3.515–6.399	0.983	0.983	0.959–1.007	0.832
	Limbs	Right front	2.010	2.007	1.803–2.214	0.523	0.523	0.504–0.540	0.793
		Left front	1.922	1.919	1.731–2.131	0.520	0.520	0.501–0.537	0.787
		Right back	6.670	6.637	5.776–7.858	0.760	0.760	0.735–0.791	0.898
		Left back	6.692	6.640	5.667–7.812	0.760	0.760	0.731–0.786	0.898
Linear Whorls	Head	Top right	0.200	0.204	0.165–0.228	0.189	0.189	0.180–0.198	0.515
		Top center	0.866	0.864	0.765–0.975	0.758	0.757	0.711–0.805	0.533
		Top left	0.402	0.407	0.337–0.457	0.270	0.270	0.258–0.287	0.598
		Between eyes	1.841	1.833	1.558–2.195	0.720	0.721	0.668–0.782	0.718
		Low right	3.991	4.094	2.419–5.175	0.748	0.742	0.453–1.055	0.842
		Low center	2.487	2.476	2.157–2.860	0.767	0.768	0.709–0.821	0.764
		Low left	3.834	3.828	3.452–4.213	0.804	0.804	0.752–0.849	0.827
	Body–neck	Right	0.136	0.136	0.131–0.140	0.079	0.077	0.075–0.081	0.636
		Center	0.136	0.136	0.131–0.140	0.078	0.079	0.075–0.807	0.636
		Left	0.102	0.102	0.097–0.107	0.128	0.129	0.125–0.132	0.442
	Limbs	Right forelimb	0.842	0.843	0.739–0.953	0.406	0.406	0.385–0.429	0.675
		Left forelimb	0.883	0.883	0.796–0.973	0.396	0.396	0.379–0.415	0.690
		Right hindlimb	4.876	4.876	4.391–5.499	0.983	0.983	0.960–1.005	0.832
		Left hindlimb	4.798	4.822	3.983–5.442	0.981	0.981	0.958–1.004	0.830

HPD 95%: 95% Highest Probability Density interval.

**Table 5 animals-13-02943-t005:** Genetic variance (σ_u_), residual variance (σ_e_) and heritability (h^2^) values for hair whorls according to their frequency of appearance at head, body–neck and limb locations.

		σ_u_			σ_e_			h^2^
Location	Particularity	Mean	Median	HPD 95%	Mean	Median	HPD 95%	
Head	Circular whorls	0.646	0.646	0.604–0.683	1.013	1.012	0.982–1.044	0.389
	Linear whorls	0.058	0.058	0.053–0.062	0.164	0.164	0.160–0.169	0.260
Body–neck	Circular whorls	1.459	1.459	1.389–1.519	1.415	1.415	1.367–1.462	0.508
	Linear whorls	1.500	1.500	1.445–1.555	1.092	1.092	1.054–1.131	0.579
Limbs	Circular whorls	0.115	0.115	0.107–0.124	0.246	0.246	0.239–0.253	0.318
	Linear whorls	0.126	0.126	0.120–0.133	0.158	0.158	0.153–0.163	0.444

HPD 95%: 95% Highest Probability Density interval.

**Table 6 animals-13-02943-t006:** Genetic correlations among linear hair whorl positions (above diagonal) and circular hair whorl positions (below diagonal).

Location and Position		Head							Body-Neck			Limbs			
		Top Right	Top Center	Top Left	Between Eyes	Low Right	Low Centre	Low Left	Right	Centre	Left	Right	Left	Right	Left
												Forelimbs		Hindlimbs	
Head	Top right		−0.081	−0.184	0.377	0.141	0.300	0.148	0.047	0.058	0.047	0.032	0.108	0.104	0.108
	Top center	0.005		0.437	0.123	0.280	0.720	0.286	0.250	0.238	0.251	−0.037	−0.054	0.156	0.152
	Top left	−0.771	0.044		0.202	0.103	0.226	0.113	0.169	0.165	0.221	−0.272	−0.206	0.085	0.055
	Between eyes	0.040	−0.440	0.081		0.266	0.306	0.290	0.380	0.379	0.382	0.095	0.129	0.152	0.147
	Low right	−0.426	0.164	0.578	−0.229		0.348	0.998	0.147	0.147	0.151	0.290	0.335	0.444	0.445
	Low center	−0.048	−0.040	−0.095	−0.446	−0.051		0.351	0.233	0.229	0.149	0.141	0.110	0.184	0.177
	Low left	−0.078	0.101	0.270	−0.195	0.807	0.053		0.146	0.146	0.146	0.279	0.323	0.443	0.443
Body-neck	Right	0.028	0.046	0.074	−0.021	0.998	−0.120	0.998		0.999	0.773	0.060	0.038	0.096	0.096
	Center	0.028	−0.017	0.062	0.220	0.239	−0.125	0.252	0.973		0.769	0.054	0.034	0.088	0.087
	Left	0.018	0.053	−0.005	0.109	0.998	−0.121	0.998	0.792	0.900		0.084	0.034	0.104	0.101
Limbs	Right forelimb	−0.172	0.007	−0.081	0.01	0.027	0.066	0.033	0.056	0.048	0.058		0.932	0.375	0.372
	Left forelimb	−0.252	0.001	0.164	0.031	0.026	0.054	0.024	0.068	0.065	0.062	0.926		0.417	0.419
	Right hindlimb	0.028	0.035	−0.024	0.030	0.038	−0.065	0.060	0.176	0.998	0.190	0.082	0.071		0.998
	Left hindlimb	0.016	0.032	−0.027	0.034	0.038	−0.055	0.048	0.998	0.998	0.189	0.096	0.061	0.828	

**Table 7 animals-13-02943-t007:** Genetic correlations between number of linear and circular whorls, according to their location on the horse.

Location		Head	Body–Neck		Limbs	
		Linear Whorls	Circular Whorls	Linear Whorls	Circular Whorls	Linear Whorls
Head	Circular whorls	0.213	0.012	−0.004	0.138	0.054
	Linear whorls		0.153	0.335	0.272	0.448
Body–neck	Circular whorls			0.429	0.349	0.130
	Linear whorls				0.183	0.125
Limbs	Circular whorls					0.618

## Data Availability

Data are unavailable due to compliance with privacy laws.

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
