# Peer review of "Phenotypic and Genetic Study of the Presence of Hair Whorls in Pura Raza Español Horses"

_animals, 2023, doi:10.3390/ani13182943_

Round 1

Reviewer 1 Report

L 20-21 "...which could ... docile horses." and L 34 "... which could ... docile horses", in my opinion both parts should be deleted, since this is not a result of this work.

L 17 (and thourough the whole manuscript) "generation interval" is misused, birth years is more appropriate.

L 117-119: birth year and inbreeding can be used as they are, I mean as covariates: authors grouped both, but they do not explain the criteria.

L 121-128: it seems to refer to a test between two groups: that's right for sex, but what about coat colour, where 4 levels are present? Unfortunately I was not able to download S1 or S2 tables. Anyway, this point must be better addressed: coat colour was in the genetic model because it is significant as an effect (e.g. an effect with 3 dof) or because some coat colour was different from other?

L 261-292 This part needs shortening: it could be also moved to the intro.

Author Response

Response to Reviewer 1 Comments

First, thank you for your time and your comments which will substantially improve our work. In the following page, I will proceed point by point to answer your comments and suggestions:

L 20-21 “…which could ... docile horses." and L “34 "... which could ... docile horses", in my opinion both parts should be deleted, since this is not a result of this work.

Authors’ response: Following your recommendation, we have rewritten these lines (L21, L34-35).

L 17 (and thourough the whole manuscript) "generation interval" is misused, birth years is more appropriate.

Authors’ response: During the preparation of this manuscript, we have given careful consideration to these aspects. Our objective was to investigate the evolution that has occurred in recent decades (periods or generation intervals) in terms of the location and number of whorls. We call these periods "generation intervals" since in the PRE, the duration of these intervals is 10 years(Perdomo-González et al., 2020 Animals 10(8), 1360). This investigation focused particularly on the period since the initiation of the breeding program for the breed in 2002. In recent years, breeders have been selecting animals based on breeding values related to morphological and functional traits relevant to Dressage, where equine behavior plays a crucial role in performance. Additionally, there has been a noticeable shift in the prevalence of coat colors within the breed, with dark colors such as bay, black, and chestnut increasing in frequency while the traditionally prevalent gray coat color has declined. These observations prompted us to examine the characteristics of animals born within the last two generational intervals, and that was the reason for making this division. This type of analysis has already been used in other studies of this breed, due to the change that has occurred in the breeders' selection system (Sánchez-Guerrero et al., 2016; Livest. Sci., 185: 148-155). Our objective has never been to analyze the specific year of birth of an animal, which is why it was not included as a regression variable. In the Material and Methods section, we have improved the explanation (see lines L123-130).Additionally, we have replaced the term "generation intervals" with "birth period" to enhance clarity and facilitate understanding. 

L 117-119: birth year and inbreeding can be used as they are, I mean as covariates: authors grouped both, but they do not explain the criteria.

Authors’ response: As with the term ‘birth period’, with the inbreeding our goal was not to include the individual inbreeding of each animal as a regression covariate. Our aim was to determine if high levels of inbreeding (equivalent to the inbreeding resulting from mating half-siblings) or very high levels (>0.1875) had an impact on the number and location of spikes and whorls. In recent studies in PRE horse, inbreeding has been considered as a fixed factor, establishing a series of classes or levels (e.g., Ripollés-Lobo et al., 2023 Ital. J. Anim. Sci. 22:407–417). Furthermore, recently studies conducted by us in this breed have shown that there is no linear relationship between inbreeding and other variables (under revision).In the Material and Methods section, we have improved the explanation (see lines L130-131).

L 121-128: it seems to refer to a test between two groups: that's right for sex, but what about coat colour, where 4 levels are present? Unfortunately I was not able to download S1 or S2 tables. Anyway, this point must be better addressed: coat colour was in the genetic model because it is significant as an effect (e.g. an effect with 3 dof) or because some coat colour was different from other?

Authors’ response: Coat color was a significant factor in 44 traits, and consequently, having one coat color or another could affect the number and position of hair whorls.

L 261-292 This part needs shortening: it could be also moved to the intro.

Authors’ response: Following your recommendation, this part has been shortened.

Reviewer 2 Report

The authors present a study of hair whorls in horses, in particular horses of the breed "pura raza española" (PRE). The authors discriminate round hair whorls and lineal hair whorls, which are longitudinal in shape.

(Actually, the latter are known from Rhodesian ridgeback dogs, where the causal mutation is known: a duplication containing FGF3 among other genes. In Rhodesian ridgebacks, ridges are inherited dominantly and associated with diseases, especially in individuals with both mutant alleles. The authors do not seem to be aware of this, at least they do not cite these studies, but this should be mentioned. If I read Table 2 correctly, the proportion of lineal whorls is low, with more than 90% of horses not having any on head and limbs, while more than half have them on body and neck. Certain rare lineal whorls could be caused by a single mutation and may be associated with diseases, as in the dogs.)

The main value of the article seems to be the large data base, which allows quite precise estimates. From Table 3, it is obvious that the number of circular whorls does not follow a Poisson distribution as one and three whorls are much more common than two and four. Counting symmetric whorls just once, may show more similarity to the theoretical Poisson distribution. In any case, these results indicate that the presence of (some) whorls is tightly associated with that of others, ie that whorls are not independent.

Generally, heritabilities of hair whorls seem quite high (Tables 4 and 5). Some correlations among whorls (especially symmetric whorls) also are high, while others are close to zero. This corresponds to the observation of associations among whorls pointed out in the previous paragraph.

While these data seem quite interesting, the article left me very confused, which I do not only attribute to the poor English language.

Examples for problems with the language are:

Legend Table 2: "due to their body location"; probably "at body locations" is meant.

Abstract: "By last" likely means "Secondly" or "Furthermore", but what is: "... were found on the between left and right positions ..."?

What are (line 329): "Light left-sided hair whorls", ie what does "light" mean in this context?

But sometimes the problem seems to run deeper than poor language:

Since a hair whorl does not per se qualify as a disease (at least the authors do not show any associations of hair whorls with diseases), it makes little sense to use the term "risk factor". Rather it is just a factor (that enters in a statistical model) that is associated with a trait. Similarly, the variable "generation interval" is redefined from its usual use in genetics, which is the interval between subsequent. Here it seems to mean the year of birth (where some years are lumped together). It is unclear, why the years are classified into 3 levels: animals born before 2001, between 2002 and 2011 and between 2012 117 and 2021. Either each year should be taken as level or the year of birth should be entered as a regression variable.

This brings me to another variable that should be entered as a regression variable: the inbreeding coefficient. I can see no sense in defining three levels and entering these as factor. I have also no clue what: "together with its influence on the number and position of hair whorls" should mean. A factor, by definition, is supposed to influence the target variable in a linear model.

Another sentence that makes no sense to me is: "In this context, horses with a gray coat color of the breed Caballo de Deporte Español (CDE), which is closely influenced by PRE horses, presented lower stress levels..." Generally, it seems that horses with gray coat were less stressed according to an infrared sensor (but note that the high reflectivity of the white coat of gray horses may influence this measurement) and the gray phenotype was associated with whorls. As associations between hair whorls and neuronally influenced traits could be postulated the authors emphasize this association. But the authors themselves call some associations between hair whorls "superstition" (paragraph starting with line 53) only to switch to scientific evidence for association between horse temper and hair whorls in the last sentence of this paragraph. Hence, I am confused what is superstition to the authors and what scientific evidence.

Similarly, I am confused reading line 357: "In general, hair whorls in the PRE horse are highly heritable, coinciding with the fact that horses born in the last generation interval (>2012-2021) present a higher proportion of these characteristics (able S1 and S2)." High heritability by definition means little influence from the environment. The time of birth is clearly an environmental influence. Hence how does the first half-sentence relate to the second?

Another sentence that left me confused is line 373: "These findings suggest that the same set of genes coordinates the position of hair whorls on one side." Do the authors mean "on both sides" here?

The next sentence is also problematic: "Hair whorls are medium-high heritability traits mainly affected by coat color, gender and generation interval and to a lesser extent by the inbreeding coefficient." While gender and time of birth (not generation interval, see above) may causally influence the formation of whorls, coat color likely correlates with hair whorls, rather than influence them causally.

Also the last sentence seems confused as two statements seem mangled: "Most horses, particularly gray ones, showed circular hair whorls below the central line of the eyes, which could be associated with calmer and more docile horses." The two statements are i) that most horses show a hair whorl below the central line of the eyes and ii) that gray horses show such a hair whorl with very high frequency. Gray horses also exhibit less stress. However, the causal arrow here is not at all clear: it could be that gray horses are more docile and by chance happen to have a higher frequency of such a hair whorl; that hair whorl and temperament are associated and that by chance many gray horses fall into this category; etc.

Generally, I consider the data set interesting but find the interpretation of results quite haphazard and confused. This confusion seems not only due to language problems.

Comments to the quality of the English language are already in the "comments and suggestions" section.

Author Response

Response to Reviewer 2 Comments

First, thank you for your time and your comments which will substantially improve our work. In the following pages, I will proceed point by point to answer your comments and suggestions:

The authors present a study of hair whorls in horses, in particular horses of the breed "pura raza española" (PRE). The authors discriminate round hair whorls and lineal hair whorls, which are longitudinal in shape.

Actually, the latter are known from Rhodesian ridgeback dogs, where the causal mutation is known: a duplication containing FGF3 among other genes. In Rhodesian ridgebacks, ridges are inherited dominantly and associated with diseases, especially in individuals with both mutant alleles. The authors do not seem to be aware of this, at least they do not cite these studies, but this should be mentioned.

Authors’ response: First of all, we apologize for not having included before this reference, as we initially carried out the bibliographical review using the term ‘whorls’. So, following your recommendation, this information has now been included (L388-L397).

If I read Table 2 correctly, the proportion of lineal whorls is low, with more than 90% of horses not having any on head and limbs, while more than half have them on body and neck. Certain rare lineal whorls could be caused by a single mutation and may be associated with diseases, as in the dogs.)

Authors’ response: There are actually very few studies that address this issue, and we were not aware of the Rhodesian ridgeback dogs study. Indeed, this could be one of the causes; for this reason,following your recommendation,we have added a sentence in the ‘Discussion’ section. (L394-397).

The main value of the article seems to be the large data base, which allows quite precise estimates. From Table 3, it is obvious that the number of circular whorls does not follow a Poisson distribution as one and three whorls are much more common than two and four. Counting symmetric whorls just once, may show more similarity to the theoretical Poisson distribution. In any case, these results indicate that the presence of (some) whorls is tightly associated with that of others, ie that whorls are not independent.

Authors’ response: Indeed, you are right, both in the statistical analysis and for estimating genetic parameters with a Bayesian model. Besides, following your recommendation this has been also discussed in the paper (L328-332).

Generally, heritabilities of hair whorls seem quite high (Tables 4 and 5). Some correlations among whorls (especially symmetric whorls) also are high, while others are close to zero. This corresponds to the observation of associations among whorls pointed out in the previous paragraph.

Authors’ response: Correct and we have already included this in the paper. The h2 are generally high (see lines 230-234) and the correlations that indicate symmetry are also high in general (see lines 398-402). In addition, other correlations that do not indicate this symmetry could have a much lower range (see lines 269-274).

While these data seem quite interesting, the article left me very confused, which I do not only attribute to the poor English language.

Authors’ response: We deeply regret this comment, as the article has been corrected and reviewed by an official English proofreading service. We have requested them to recheck it and provide us with a certificate as well.

Legend Table 2: "due to their body location"; probably "at body locations" is meant.

Authors’ response: We think that you are talking about Table 4, and following your recommendation, this has been changed (L237).

Abstract: "By last" likely means "Secondly" or "Furthermore", but what is: "... were found on the between left and right positions ..."?

Authors’ response: Following your recommendation, these phrases have been changed (L26).

What are (line 329): "Light left-sided hair whorls", ie what does "light" mean in this context?

Authors’ response: Following your recommendation, we have explainedthis better (L345)

Since a hair whorl does not per se qualify as a disease (at least the authors do not show any associations of hair whorls with diseases), it makes little sense to use the term "risk factor". Rather it is just a factor (that enters in a statistical model) that is associated with a trait.

Authors’ response: Following your recommendation, this term has been changed (L27, L117-118, L148-149,L209).

Similarly, the variable "generation interval" is redefined from its usual use in genetics, which is the interval between subsequent. Here it seems to mean the year of birth (where some years are lumped together). It is unclear, why the years are classified into 3 levels: animals born before 2001, between 2002 and 2011 and between 2012 117 and 2021. Either each year should be taken as level or the year of birth should be entered as a regression variable.

Authors’ response: During the preparation of this manuscript, we have given careful consideration to these aspects. Our objective was to investigate the evolution that has occurred in recent decades (periods or generation intervals) in terms of the location and number of whorls. We call these periods "generation intervals" since in the PRE, the duration of these intervals is 10 years(Perdomo-González et al., 2020 Animals 10(8), 1360). This investigation focused particularly on the period since the initiation of the breeding program for the breed in 2002. In recent years, breeders have been selecting animals based on breeding values related to morphological and functional traits relevant to Dressage, where equine behavior plays a crucial role in performance. Additionally, there has been a noticeable shift in the prevalence of coat colors within the breed, with dark colors such as bay, black, and chestnut increasing in frequency while the traditionally prevalent gray coat color has declined. These observations prompted us to examine the characteristics of animals born within the last two generational intervals, and that was the reason for making this division. This type of analysis has already been used in other studies of this breed, due to the change that has occurred in the breeders' selection system (Sánchez-Guerrero et al., 2016; Livest. Sci., 185: 148-155). Our objective has never been to analyze the specific year of birth of an animal, which is why it was not included as a regression variable. In the Material and Methods section, we have improved the explanation (see lines L123-130).Additionally, we have replaced the term "generation intervals" with "birth period" to enhance clarity and facilitate understanding.

This brings me to another variable that should be entered as a regression variable: the inbreeding coefficient. I can see no sense in defining three levels and entering these as factor. I have also no clue what: "together with its influence on the number and position of hair whorls" should mean. A factor, by definition, is supposed to influence the target variable in a linear model.

Authors’ response: As with the term ‘birth period’, with the inbreeding our goal was not to include the individual inbreeding of each animal as a regression covariate. Our aim was to determine if high levels of inbreeding (equivalent to the inbreeding resulting from mating half-siblings) or very high levels (>0.1875) had an impact on the number and location of spikes and whorls. In recent studies in PRE horse, inbreeding has been considered as a fixed factor, establishing a series of classes or levels (e.g., Ripollés-Lobo et al., 2023 Ital. J. Anim. Sci. 22:407–417). Furthermore, recently studies conducted by us in this breed have shown that there is no linear relationship between inbreeding and other variables (under revision).In the Material and Methods section, we have improved the explanation (see lines L130-131).

Another sentence that makes no sense to me is: "In this context, horses with a gray coat color of the breed Caballo de Deporte Español (CDE), which is closely influenced by PRE horses, presented lower stress levels..." Generally, it seems that horses with gray coat were less stressed according to an infrared sensor (but note that the high reflectivity of the white coat of gray horses may influence this measurement) and the gray phenotype was associated with whorls.

Authors’ response: We understand the Reviewer’sconcerns about stress measurement in CDE horses; however, these infrared thermography measurements were assessed on the caruncle of the eye, measuring the temperature of the blood flow in this area since, as reported in this paper, it appeared to be related with an Autonomous Nervous System response and thus, with a stress response. Thus, no bias on the results should be considered here due to the reflectivity of the white coat of gray horses, since the measurements were not obtained on the animal’s hair, as the Reviewer suggests. As regards our postulated association between hair whorls and neuronally influenced traits, this sentence has been changed, as we have presented no proof for that in this manuscript.

As associations between hair whorls and neuronally influenced traits could be postulated the authors emphasize this association. But the authors themselves call some associations between hair whorls "superstition" (paragraph starting with line 53) only to switch to scientific evidence for association between horse temper and hair whorls in the last sentence of this paragraph. Hence, I am confused what is superstition to the authors and what scientific evidence.

Authors’ response: The association of hair whorls with “superstition” was made in the Introduction section, as a way of describing the history and evolution of the study of this effect, which started with superstitions and popular beliefs and then progressed to sound scientific and experimental studies which could prove this relation. However, following the Reviewer’s suggestions, we have rephrased the paragraph in order to clarify it. (L62-64).

Similarly, I am confused reading line 357: "In general, hair whorls in the PRE horse are highly heritable, coinciding with the fact that horses born in the last generation interval (>2012-2021) present a higher proportion of these characteristics (able S1 and S2)." High heritability by definition means little influence from the environment. The time of birth is clearly an environmental influence. Hence how does the first half-sentence relate to the second?

Authors’ response: Indeed, whorls are highly heritable, and as a result, the number of animals exhibiting them increases over time. As we mentioned in previous responses, our objective was not to analyze the year of birth of the animal, but rather the birth period that corresponds to a specific period in the breed's selection process. It has been observed, perhaps due to indirect selection (seeking animals with a specific behavior and coat color), that the number of whorls has increased in the last generational interval analyzed. However, we have attempted to clarifyour explanation of this in the document (see lines 375-379).

Another sentence that left me confused is line 373: "These findings suggest that the same set of genes coordinates the position of hair whorls on one side." Do the authors mean "on both sides" here?

Authors’ response: Following your suggestion,this has been rewritten (L403).

The next sentence is also problematic: "Hair whorls are medium-high heritability traits mainly affected by coat color, gender and generation interval and to a lesser extent by the inbreeding coefficient." While gender and time of birth (not generation interval, see above) may causally influence the formation of whorls, coat color likely correlates with hair whorls, rather than influence them causally.

Authors’ response: According to the Reviewer’s suggestions, this paragraph has been divided into two statements and clarified (L406-408, L412-416).

Also the last sentence seems confused as two statements seem mangled: "Most horses, particularly gray ones, showed circular hair whorls below the central line of the eyes, which could be associated with calmer and more docile horses." The two statements are i) that most horses show a hair whorl below the central line of the eyes and ii) that gray horses show such a hair whorl with very high frequency. Gray horses also exhibit less stress. However, the causal arrow here is not at all clear: it could be that gray horses are more docile and by chance happen to have a higher frequency of such a hair whorl; that hair whorl and temperament are associated and that by chance many gray horses fall into this category; etc.

Authors’ response: According to the Reviewer’s suggestions, this paragraph has been divided into two statements and clarified (L406-408, L412-416).

Generally, I consider the data set interesting but find the interpretation of results quite haphazard and confused.Thisconfusionseemsnotonlydue to languageproblems.

Authors’ response: Indeed, the amount of data and number of analyses used in this work have been very high. Although we have added some supplementary tables, the volume of results generated has forced us to make the tables large, which can make them more difficult to read.Following your suggestions here, we have made an effort to improve all the paper, and have had the English checked for a second time.

Round 2

Reviewer 1 Report

The submission has been substantially improved, but I must confirm my doubts about 2.2 Statistical analysis.

There are two group of traits in the manuscript: dichotomous (presence|absence in a position) and grades (number of whorls); furthermore, there are fixed effect with two levels (sex), or effects with more than two levels (year period, coat and inbreeding). The significance levels has been estimated by z. This method is appropriate only for the effect of sex (two levels) on dichotomous trait. Extending the method to 4*3/2 paired comparisons, like authors did for coat colors, greatly increase the overall type I error rate; things go worse if the comparison refers to each grade of the scale .

Author Response

Reviewer 1

First and foremost, we express our sincere gratitude for your comments, which have significantly enhanced the paper and its comprehensibility.

We would like to sincerely apologize for not being able to adequately explain the statistical analyses conducted in our research. We acknowledge that our initial explanation and methodology of the statistical methods were not as clear as they should have been.

We have taken your comments and feedback seriously and have made diligent efforts to provide a much clearer explanation and methodology of the statistical analyses in the revised paper. Your valuable insights have significantly contributed to the improvement of our work and its overall understanding.

Once again, we express our gratitude for your valuable feedback and the time you have taken to review our paper. We believe that the revisions we have made will enhance the quality and rigor of our research.

Thank you very much for your understanding and support.

There are two group of traits in the manuscript: dichotomous (presence|absence in a position) and grades (number of whorls); furthermore, there are fixed effect with two levels (sex), or effects with more than two levels (year period, coat and inbreeding). The significance levels has been estimated by z. This method is appropriate only for the effect of sex (two levels) on dichotomous trait. Extending the method to 4*3/2 paired comparisons, like authors did for coat colors, greatly increase the overall type I error rate; things go worse if the comparison refers to each grade of the scale.

ANS: We have made improvements to the explanation and methodology of the statistical analyses (L118-120). The p-values for each risk factor were exclusively calculated using the GLZ method, and in tables S1 and S2, we have now included a column displaying the corresponding GLZ p-values.

For dichotomous traits, we utilized a binomial distribution, while for discrete traits, a multinomial distribution was employed, as comprehensively described in the paper (L118-210).

Regarding the comparison of frequencies in tables S1 and S2, we performed a z-test, specifically applying a two-by-two comparison. This method was not deemed appropriate since it is applicable when the factor under consideration has only two levels. Consequently, to maintain clarity, we have retained the frequencies in tables S1 and S2 while eliminating the statistical comparison test.

Reviewer 2 Report

The manuscript is generally much easier to read now; the language is improved and many of the issues I raised turned out to be due to awkward language rather than genuine problems. There seem to be spaces missing between new and old parts, likely a technical problem.

I would still have preferred that the inbreeding coefficient would not enter as factor but as a continuous variable. I also had a closer look at the statistics, which I did not do previously due to general confusion. The authors claim to use a generalized non-linear model (which they abbreviate with GLZ, while eg GNLM would seem more intuitive). I cannot see, however, what the non-linear part in the model is. Furthermore, the authors write: "The test used to compute the significance level for the difference between each two proportions was...". This comment actually suggests a generalized linear model of the binomial family, which allows for analysis of proportions. The following list is, however, not really an explanation for which model was run (as claimed), but a list of variables that enter the model. The explanation how the p-value was calculated seems to be a transformation to a normal distribution. With generalized linear models such a transformation is not done and p-values are calculated differently than in line 40. Later they use BLUP (which is the best LINEAR unbiased predictor), ie, a linear but not a non-linear model, and report p-values. The authors further mix in a Bayesian approach, which they claim they used universally: "All the models were analyzed using a Bayesian approach via Gibbs sampling with 153 the GIBBSF90+ module of the BLUPF90 software [38]." But it seems that many analyses are run within the frequentist paradigm as p-values are reported (see above). Hence, I am (as during my lprevious review) confused; this time about statistics. To make it plain: the description of the statistical methods is so unclear that I am not convinced that analyses are correct. I would hope that this can be resolved as well.

If I assume that the results hold, I can follow the arguments from beginning till the end.

The English is much improved and generally OK.

Author Response

Reviewer 2

First and foremost, we express our sincere gratitude for your comments, which have significantly enhanced the paper and its comprehensibility.

The manuscript is generally much easier to read now; the language is improved and many of the issues I raised turned out to be due to awkward language rather than genuine problems. There seem to be spaces missing between new and old parts, likely a technical problem.

Ans: The authors has improved the spaces missing.

I would still have preferred that the inbreeding coefficient would not enter as factor but as a continuous variable

Ans: As we mentioned earlier, we have thoroughly considered this matter, and it is crucial for the coherence and context of this paper to maintain the inbreeding categorization into classes.

I also had a closer look at the statistics, which I did not do previously due to general confusion. The authors claim to use a generalized non-linear model (which they abbreviate with GLZ, while eg GNLM would seem more intuitive). I cannot see, however, what the non-linear part in the model is.

Ans: The generalized non-linear model represents a methodological approach analogous to a GLM, with the key distinction being the absence of a continuous Gaussian distribution assumption. Instead, the model assumes other distributions based on the nature of the data being analyzed. In our study, the GLZ (commonly referred to as such) has been applied to dichotomic data, wherein it has been fitted to a binomial logit distribution. Moreover, the variables pertaining to the number of circular whorls and linear whorls have been fitted to a multinomial ordinal logit distribution. We have incorporated this updated information in the paper (L118-210) to facilitate comprehension and clarity for readers. Besides, the tables S1 and S2 have been modified.

The authors write: "The test used to compute the significance level for the difference between each two proportions was...". This comment actually suggests a generalized linear model of the binomial family, which allows for analysis of proportions.

Ans: We have made improvements to the explanation and methodology of the statistical analyses (L118-120). The p-values for each risk factor were exclusively calculated using the GLZ method, and in tables S1 and S2, we have now included a column displaying the corresponding GLZ p-values. For dichotomous traits, we utilized a binomial distribution, while for discrete traits, a multinomial distribution was employed, as comprehensively described in the paper (L118-210). Regarding the comparison of frequencies in tables S1 and S2, we performed a z-test, specifically applying a two-by-two comparison. This method was not deemed appropriate since it is applicable when the factor under consideration has only two levels. Consequently, to maintain clarity, we have retained the frequencies in tables S1 and S2 while eliminating the statistical comparison test.

The following list is, however, not really an explanation for which model was run (as claimed), but a list of variables that enter the model. The explanation how the p-value was calculated seems to be a transformation to a normal distribution. With generalized linear models such a transformation is not done and p-values are calculated differently than in line 40.

Ans: clarified in the previous answer.

Later they use BLUP (which is the best LINEAR unbiased predictor), ie, a linear but not a non-linear model, and report p-values.

Ans: The p-values were computed solely through GLZ analysis, serving as a preliminary measure to ascertain the significance or non-significance of the proposed effects and to determine the effects to be incorporated into the genetic model. The BLUP model (general terminology), employing a Bayesian approach, was utilized for estimating genetic parameters, as it does not necessitate data adherence to any specific distribution. To avoid confusion, BLUP has been deleted from the document.

The authors further mix in a Bayesian approach, which they claim they used universally: "All the models were analyzed using a Bayesian approach via Gibbs sampling with 153 the GIBBSF90+ module of the BLUPF90 software [38]." But it seems that many analyses are run within the frequentist paradigm as p-values are reported (see above).

Ans: See previous comment. Clarified in the text (L137).

Hence, I am (as during my lprevious review) confused; this time about statistics. To make it plain: the description of the statistical methods is so unclear that I am not convinced that analyses are correct. I would hope that this can be resolved as well.

Ans: I would like to sincerely apologize for not being able to adequately explain the statistical analyses conducted in our research. We acknowledge that our initial explanation and methodology of the statistical methods were not as clear as they should have been.

We have taken your comments and feedback seriously and have made diligent efforts to provide a much clearer explanation and methodology of the statistical analyses in the revised paper. Your valuable insights have significantly contributed to the improvement of our work and its overall understanding.

Once again, we express our gratitude for your valuable feedback and the time you have taken to review our paper. We believe that the revisions we have made will enhance the quality and rigor of our research.

Thank you very much for your understanding and support.

If I assume that the results hold, I can follow the arguments from beginning till the end.

Round 3

Reviewer 1 Report

Now the statistical methods are adequately explained and I have no further remarks.

Author Response

Dear reviewer:

Thank you so much. 

Reviewer 2 Report

Both in the article as well as in the response to my criticism the authors are so vague and imprecise that I cannot really tell which method was used. For dichotomous response variables a generalized linear model (usually abbreviated as GLM) of the binomial family with logit link function seems to be assumed and probably ordinal regression in another. While the link function is assumed to be nonlinear (eg logit), the explanatory variable seems to be assumed to be linear. For comparison, eg a quadratic regression would be non-linear, even though normally distributed data (and thus a linear link function) may be assumed. The term "linear" in "generalized linear model" does not refer to the link function but to linearity in response. An easily accessible and generally correct text on GLMs is:

https://en.wikipedia.org/wiki/Generalized_linear_model

In particular, the following sentence makes no sense: "A binomial logit distribution using GLZ was used for the dichotomic traits, while for discrete traits it was used a GLZ with an ordinal multinomial distribution." I guess what is meant is: "A generalized linear model of the binomial family with a logit link function was used for counts of dichotomous traits." Even though I am teaching statistics and have worked as a statistical consultant for years, I have not yet encountered an "ordinal multinomial distribution". The authors may mean an ordinal regression model or a cumulative logit (or probit) model (see refs below, further info and refs also: https://faculty.washington.edu/heagerty/Courses/b571/handouts/MultModels.pdf; look also at chapter 11.6 of the R-tutorial for simple models: https://cran.r-project.org/doc/manuals/r-release/R-intro.html). If the authors would have cited the statistical package and function(s) they used, I could have guessed more.

In any case, if indeed such a complicated model was used, more information and references should have been provided. If there is another convention that corresponds to the authors use of language, many more references would be needed.

The response to my other comments was rather minimal. I recommend that the authors try again more carefully.

Agresti, Alan. 2002. Categorical Data Analysis. A Wiley-Interscience Publication. Wiley.

Betancourt, Michael. 2019. “Ordinal Regression.” https://betanalpha.github.io/assets/case

_studies/ordinal_regression.html.

Bürkner, Paul-Christian, and Matti Vuorre. 2019. “Ordinal Regression Models in Psychology:

A Tutorial.” Advances in Methods and Practices in Psychological Science 2 (1): 77–101.

https://doi.org/10.1177/2515245918823199.

Improved compared to the first version

Author Response

Dear reviewer,

Both in the article as well as in the response to my criticism the authors are so vague and imprecise that I cannot really tell which method was used.

ANS: We deeply appreciate the time and effort you have devoted to reviewing our manuscript. Throughout each revision, we have conscientiously endeavored to clarify and enhance the manuscript in accordance with your guidance. We trust that these clarifications now align with your expectations.

In our first versions, we consider that delving into a type of complex statistical analysis, such as the GLZ, does not make much sense for the average reader of this type of work, especially when it is a preliminary analysis to select the significates effects to be included in the specific statistical-genetic analysis to estimation of genetic parameters). However, since the reviewer considers the information to be extensive, we have completed the description of this preliminary statistical analysis (lines 118-122).

For dichotomous response variables a generalized linear model (usually abbreviated as GLM) of the binomial family with logit link function seems to be assumed and probably ordinal regression in another. While the link function is assumed to be nonlinear (eg logit), the explanatory variable seems to be assumed to be linear. For comparison, eg a quadratic regression would be non-linear, even though normally distributed data (and thus a linear link function) may be assumed. The term "linear" in "generalized linear model" does not refer to the link function but to linearity in response

ANS: Given the characteristics of our dependent variables, we have used the Generalized Linear Model (GLZ), in no case the GLM, to determine if the risk factors evaluated have a statistically significant impact on the dependent variables. GLZ is a generalization of the general linear model, which is used when the dependent variable of interest has a non-continuous distribution or a continuous non-normal distribution (and thus, the predicted values should also follow the respective distribution). In our case we have assumed a binomial distribution for the dichotomous variables and an ordinal multinomial (a multinomial discrete distribution containing information on a ranks scale) for the multinomial ones (for the count of whorls -ranging from 0 to 6-) GLZ uses the maximum likelihood methods to build models to estimate and test hypotheses about effects in the model.

Various link functions linking the dependent and predictor variables can be chosen, contingent upon the assumed distribution of the dependent variable values (see McCullagh & Nelder, 1989. Generalized linear models. 2nd Ed. New York: Chapman & Hall). We opted for the Logit link function: f(z) = log(z/(1-z)), utilizing a factorial between-subject design (equivalent to a factorial ANOVA of GLM) to ascertain the statistical significance of effects on the dependent variable.

For an in-depth exploration of generalized linear models, we refer readers to:

  • Dobson, 1990. An introduction to generalized linear models. New York: Chapman & Hall.
  • Green & Silverman, 1994. Nonparametric regression and generalized linear models: A roughness penalty approach. New York: Chapman & Hall.
  • McCullagh & Nelder, 1989. Generalized linear models (2nd Ed.). New York: Chapman & Hall.

As well as the implementation details found within used Statistical software: https://statisticasoftware.wordpress.com/2012/07/16/generalized-linear-models-glz-statistics/

Finally, as an example of other works where this type of statistical procedure is used, we can recommend Soler et al., 2009 (doi: 10.1186/1471-2148-9-88) or Schradin et al. 2009 (doi.org/10.1095/biolreprod.108.075838).

In particular, the following sentence makes no sense: "A binomial logit distribution using GLZ was used for the dichotomic traits, while for discrete traits it was used a GLZ with an ordinal multinomial distribution." I guess what is meant is: "A generalized linear model of the binomial family with a logit link function was used for counts of dichotomous traits." Even though I am teaching statistics and have worked as a statistical consultant for years, I have not yet encountered an "ordinal multinomial distribution". The authors may mean an ordinal regression model or a cumulative logit (or probit) model (see refs below, further info and refs also: https://faculty.washington.edu/heagerty/Courses/b571/handouts/MultModels.pdf; look also at chapter 11.6 of the R-tutorial for simple models: https://cran.r-project.org/doc/manuals/r-release/R-intro.html). If the authors would have cited the statistical package and function(s) they used, I could have guessed more.

ANS: We hope that the aforementioned phrase makes sense to the reviewer after the previous answer. The software used (Statistics for Windows software v.11) is cited since the first paper version in line 130 (137 in last version).

Round 4

Reviewer 2 Report

By now, the authors and I seem to agree on the methods, but still not entirely on the acronyms, labels, and terms. We agree that a generalized (non-)linear model was used, which the authors abbreviate as GLZ (to emphasize the 'iZed'). While I have not come across this abbreviation before reading this article, an internet search showed me that others also use it. Indeed it makes sense to use the 'Z' to differentiate between a General and a GeneraliZed Linear Model (without and with (non-linear) link function, respectively). Hence, the authors convinced me that using this abbreviation for a Generalized Linear Model makes sense.

I still have trouble though with describing the method as Generalized Non-linear Model. Here we need to differentiate between the explanatory/independent variables and the dependent variable. I would use the term "generalized linear model of the binomial family with logit link function" for (some of) the authors' analyses, because the explanatory variables enter the model as factors as in any linear model, even though the link function (in this case the logit function) that transforms an underlying variable is non-linear. I would reserve "non-linear" in a generalized XX model to independent variables that enter non-linearly, eg a quadratic term in a regression. The authors, on the other hand, use generalized non-linear model to refer to the non-linearity of the link function. I consider this misleading and suggest it should be changed (sent via e-mail).

Also I have not come across the term: "ordinal multinomial", which the authors explain in the response letter as "a multinomial discrete distribution containing information on a rank scale". In this case, I guess it is enough to add this explanation in parentheses the first time the term comes up (sent via e-mail).

An aside: I thought I could get a clue as to which method was used from the name of the function the authors use. I am a user of the statistical programming language "R"; the use of the function "glm(..., family = binomial(link = logit),...)" would have clued me in to the method used. (Notice, as an aside, that the very widely used "R" also uses the abbreviation "GLM" and not "GLZ".) But obviously the authors use "Statistics for Windows" and I could not get such a clue.

I guess that most readers of "Animals" will care little about this discussion.

Author Response

Dear reviewer,

By now, the authors and I seem to agree on the methods, but still not entirely on the acronyms, labels, and terms. We agree that a generalized (non-)linear model was used, which the authors abbreviate as GLZ (to emphasize the 'iZed'). While I have not come across this abbreviation before reading this article, an internet search showed me that others also use it. Indeed it makes sense to use the 'Z' to differentiate between a General and a GeneraliZed Linear Model (without and with (non-linear) link function, respectively). Hence, the authors convinced me that using this abbreviation for a Generalized Linear Model makes sense.

I still have trouble though with describing the method as Generalized Non-linear Model. Here we need to differentiate between the explanatory/independent variables and the dependent variable. I would use the term "generalized linear model of the binomial family with logit link function" for (some of) the authors' analyses, because the explanatory variables enter the model as factors as in any linear model, even though the link function (in this case the logit function) that transforms an underlying variable is non-linear. I would reserve "non-linear" in a generalized XX model to independent variables that enter non-linearly, eg a quadratic term in a regression. The authors, on the other hand, use generalized non-linear model to refer to the non-linearity of the link function. I consider this misleading and suggest it should be changed (sent via e-mail).

Also I have not come across the term: "ordinal multinomial", which the authors explain in the response letter as "a multinomial discrete distribution containing information on a rank scale". In this case, I guess it is enough to add this explanation in parentheses the first time the term comes up (sent via e-mail).

An aside: I thought I could get a clue as to which method was used from the name of the function the authors use. I am a user of the statistical programming language "R"; the use of the function "glm(..., family = binomial(link = logit),...)" would have clued me in to the method used. (Notice, as an aside, that the very widely used "R" also uses the abbreviation "GLM" and not "GLZ".) But obviously the authors use "Statistics for Windows" and I could not get such a clue.

I guess that most readers of "Animals" will care little about this discussion.

Answer: Thank you very much for all contributions. The paper has been modified according to them (they can be found marked in red).
